# CaTok: Taming Mean Flows for One-Dimensional Causal Image Tokenization

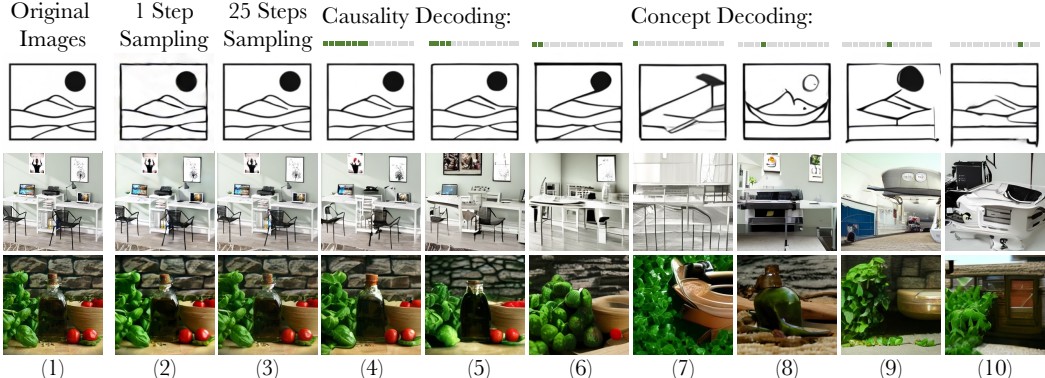

Figure 1: **Reconstruction samples.** CaTok with a MeanFlow decoder (Geng et al., 2025) supports fast one-step (col. 2) and high-quality multi-step (col. 3) sampling with 256 tokens. Reconstructions in cols. 3–7 show a fine-to-coarse trend as tokens are reduced from 256 to 16, highlighting the causality of the 1D tokens. Cols. 7–10 present reconstructions from different 16-token segments, demonstrating that CaTok naturally learns diverse visual concepts across token intervals.

## Abstract

Autoregressive (AR) language models rely on causal tokenization, but extending this paradigm to vision remains non-trivial. Current visual tokenizers either flatten 2D patches into non-causal sequences or enforce heuristic orderings that misalign with the "next-token prediction" pattern. Recent diffusion autoencoders similarly fall short: conditioning the decoder on all tokens lacks causality, while applying nested dropout mechanism introduces imbalance. To address these challenges, we present CaTok, a 1D causal image tokenizer with a MeanFlow decoder. By selecting tokens over time intervals and binding them to the MeanFlow objective, as illustrated in Fig. 1, CaTok learns causal 1D representations that support both fast one-step generation and high-fidelity multi-step sampling, while naturally capturing diverse visual concepts across token intervals. To further stabilize and accelerate training, we propose a straightforward regularization REPA-A, which aligns encoder features with Vision Foundation Models (VFMs). Experiments demonstrate that CaTok achieves state-of-the-art results on ImageNet reconstruction, reaching 22.72 PSNR and 0.681 SSIM with fewer training epochs, and the AR model attains performance comparable to leading approaches.

## 1 Introduction

The autoregressive (AR) paradigm enables generative large language models (LLMs) to achieve remarkable progress, exhibiting strong generalization and scalability (Achiam et al., 2023; OpenAI, 2025; Comanici et al., 2025; Grattafiori et al., 2024; Yang et al., 2025; Liu et al., 2024). Following the natural reading order of the text, LLMs tokenize a sentence into 1D causal tokens and perform generative modeling through next-token prediction. To emulate the capabilities and properties of LLMs in visual generation, the computer vision community has recently advanced large autoregressive

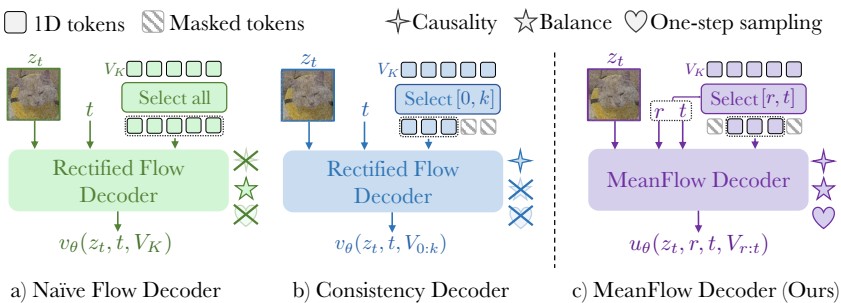

Figure 2: **Comparison among different decoders. a)** Naïve flow decoders (Sargent et al., 2025) condition on all 1D tokens from the encoder without dropout, leading the 1D tokens to lack causality; **b)** Consistency decoders obtain $k$ by random sampling (Bachmann et al., 2025; Wen et al., 2025) or timestep binding (Pan et al., 2025; Wang et al., 2025a), and condition on the first $k$ 1D tokens, which biases toward early tokens, introducing imbalance, leading to degraded performance of AR generation; **c)** Our MeanFlow decoder conditions on 1D tokens within the time interval $[r, t]$ to model the average velocity field along the subpath, which inherently maintains **causality** and **balance** of the 1D tokens, and supporting **one-step sampling** during reconstruction or generation.

vision models (Bai et al., 2024; Yu et al., 2024a; Wang et al., 2024; Sun et al., 2024; Han et al., 2025; Wang et al., 2025b). However, due to inferior performance, diffusion-based models (Ho et al., 2020; Song et al., 2021) like rectified flows (Liu et al., 2022; Lipman et al., 2022) remain the dominant approach in most scenarios (OpenAI, 2024; Wu et al., 2025).

In this paper, we argue that a crucial step toward bridging the gap between autoregressive language models and vision models lies in the causal tokenization of visual content. Autoregressive modeling relies on causal tokens and requires a predefined order of data. Unlike text, which inherently possesses a natural order, defining an appropriate order for images remains an open issue. VQGAN-like models (Esser et al., 2021) tokenize an image into grids of 2D tokens, and flatten them to a 1D sequence in raster (Razavi et al., 2019; Ramesh et al., 2021) or random (Yu et al., 2024b; Li et al., 2024b) order, which lacks causality between preceding and succeeding tokens (Pan et al., 2025; Wang et al., 2025a). VAR-like models (Tian et al., 2024; Han et al., 2025), on the other hand, tokenize images into multi-scale 2D tokens and establish a coarse-to-fine ordering via next-scale prediction. While this approach guarantees causality in visual tokens and yields promising results, it compromises the "next-token prediction" pattern of LLMs.

With the recent advances in 1D tokenizers (Cui et al., 2024; Yu et al., 2024c), the community has renewed its interest in diffusion autoencoders (Preechakul et al., 2022; Yang & Mandt, 2023) due to their demonstrated effectiveness in visual generation. Diffusion autoencoders extract 1D tokens from registers (Darcet et al., 2024) of encoders, and use them as conditions for the decoder to reconstruct images with denoising or rectified flow objective. However, as shown in Fig. 2 a), Naïve flow decoders, such as FlowMo (Sargent et al., 2025), condition on all 1D tokens from the encoder, causing the 1D tokens to lack causality and making AR learning difficult. To learn the causality for 1D tokens, as shown in Fig. 2 b), consistency decoders apply nested dropout (Rippel et al., 2014) by conditioning on the first $k$ tokens, where $k$ is determined either via random sampling, as in FlexTok (Bachmann et al., 2025) and Semanticist (Wen et al., 2025), or via timestep binding, as in DDT (Pan et al., 2025) and Selftok (Wang et al., 2025a). Since earlier tokens are more likely to be selected, this approach introduces imbalance and can be harmful to AR generation (see Tab. 3b).

Motivated by these observations, we propose CATOK, a 1D **CA**usal image **TOK**enizer equipped with a MeanFlow decoder (Geng et al., 2025). As illustrated in Fig. 2 c), we address the imbalance problem by selecting 1D tokens within a sampled time interval $[r, t]$ and binding them with the corresponding time interval in the MeanFlow objective. This allows the 1D tokens to model the average velocity field along the subpath from $r$ to $t$, capturing causality in the noise-to-image generation process while naturally supporting one-step sampling during generation. Moreover, inspired by REPA and REPA-E (Yu et al., 2024d; Leng et al., 2025), we align the image features from encoders with high-quality external visual representations, providing a regularization that effectively accelerates and stabilizes autoencoder training. We refer to this variant as REPA-A.

As shown in Fig. 1, CᴀTᴏᴋ supports both fast one-step sampling (col. 2) and high-quality multi-step sampling (col. 3) with 256 tokens, demonstrating its flexibility in balancing efficiency and fidelity. Reconstructions in cols. 3–7, obtained by progressively reducing the number of tokens from 256 to 16, exhibit a clear fine-to-coarse trend, providing evidence for the causality of the learned 1D tokens. Moreover, reconstructions in cols. 7–10 from different 16-token segments show that CᴀTᴏᴋ naturally learns diverse visual concepts across token intervals, underscoring its ability to disentangle semantic information and distribute it meaningfully among tokens. Our contributions can be summarized as:

1. We propose a novel architecture for 1D causal image tokenization based on diffusion autoencoders (Preechakul et al., 2022) with the MeanFlow (Geng et al., 2025) objective.

2. We seamlessly combine the training of a causal encoder and a one-step flow decoder, enabling one-step sampling in diffusion autoencoders *for the first time*.

3. We propose REPA-A, an advanced technique that leverages existing vision foundation models to stabilize and accelerate diffusion autoencoder training.

4. On ImageNet, our CᴀTᴏᴋ-L achieves state-of-the-art results with 22.72 PSNR and 0.681 SSIM, while attains comparable performance o leading approaches with 0.84 rFID and 3.40 gFID.

## 2 BACKGROUND

In this section, we provide a concise introduction to rectified flows (Liu et al., 2022; Lipman et al., 2022) and MeanFlow models (Geng et al., 2025) as a preliminary to our CᴀTᴏᴋ.

### 2.1 RECTIFIED FLOWS

Given data $x \sim p_{data}(x)$ and prior $\epsilon \sim p_{prior}(\epsilon)$, rectified flows learn the conditional velocity fields $v_t = v_t(z_t|x)$ between these two distributions. Specifically, a flow path can be constructed as $z_t = (1-t)x + t\epsilon$ with time $t$, and the conditional velocity can be derived by:

$$v(z_t|x) = \frac{d}{dt}z_t = \epsilon - x. \tag{1}$$

A deep neural network $v_\theta(z_t, t)$ parameterized by $\theta$ is learned to model the marginal velocity field

$$v(z_t, t) \triangleq \mathbb{E}_{p_t(v_t|z_t)}[v_t], \tag{2}$$

which is equivalent to fitting the conditional velocity field in Eq. (1) (Lipman et al., 2022). In inference, starting from $z_1 = \epsilon \sim p_{prior}(\epsilon)$, samples can be generated by solving:

$$z_r = z_t - \int_r^t v_\theta(z_\tau, \tau)d\tau, \tag{3}$$

where $r$ denotes another timestep and $r < t$. In practice, this integral is numerically approximated in discrete time steps. For instance, the Euler method updates each step as:

$$z_r = z_t - (t-r)v_\theta(z_t, t). \tag{4}$$

However, it estimates the average velocity over the interval $[r, t]$ using only the instantaneous velocity at time $t$, which introduces inaccuracies during sampling.

### 2.2 MEANFLOW MODELS

To mitigate the errors that arise with fewer sampling steps, MeanFlow models directly fit the average velocity $u$ over the interval $[r, t]$. Formally, the average velocity $u$ can be defined as:

$$u(z_t, r, t) \triangleq \frac{1}{t-r} \int_r^t v(z_\tau, \tau)d\tau. \tag{5}$$

Through derivations in Geng et al. (2025), the average velocity $u(z_t, r, t)$ can be obtained from the instantaneous velocity $v(z_t, t)$:

$$u(z_t, r, t) = v(z_t, t) - (t-r)(v(z_t, t)\partial_z u(z_t, r, t) + \partial_t u(z_t, r, t)), \tag{6}$$

and the MeanFlow objective is:

$$\mathcal{L}(\theta) = \mathbb{E}||u_\theta(z_t, r, t) - \text{sg}[v(z_t|x) - (t-r)(v(z_t|x)\partial_z u_\theta + \partial_t u_\theta)]||_2^2, \tag{7}$$

where sg[·] denotes the stop-gradient operation, avoiding double backpropagation through the Jacobian–vector product. Moreover, one-step sampling can be given by $z_0 = \epsilon - u_\theta(\epsilon, 0, 1)$.

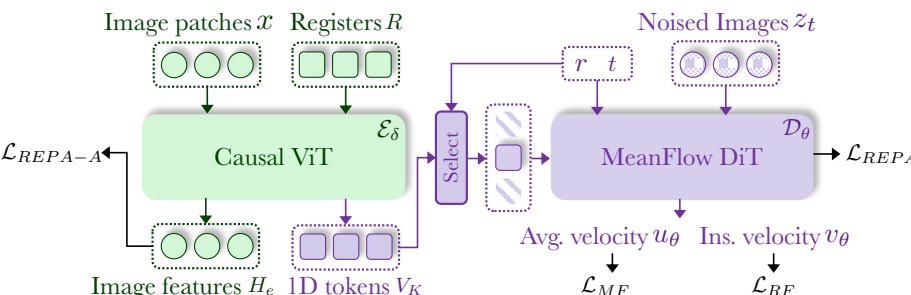

Figure 3: **Architecture of our CATOK.** CATOK is a diffusion autoencoder with a causal Vision Transformer (ViT) (Dosovitskiy et al., 2021) encoder and a MeanFlow Diffusion Transformer (DiT) (Peebles & Xie, 2023) decoder. The encoder leverages registers (Darcet et al., 2024) to extract rich visual information into 1D tokens, which are then conditioned to the decoder through time interval selecting. With two flow objectives and two representation alignment objectives, CATOK effectively learns causal 1D representations that support both one-step and multi-step sampling, while naturally capturing diverse visual concepts across different token intervals.

## 3 CATOK

We now introduce CATOK, a diffusion autoencoder (Preechakul et al., 2022; Yang & Mandt, 2023) with a causal Vision Transformer (ViT) (Dosovitskiy et al., 2021) encoder and a MeanFlow Diffusion Transformer (DiT) (Peebles & Xie, 2023) decoder, for 1D causal image tokenization. We begin in Sec. 3.1 by presenting the architecture of CATOK. Next, in Sec. 3.2, we describe how it is optimized through multiple objectives. Finally, in Sec. 3.3, we outline the autoregressive modeling procedure used for image generation with the trained CATOK.

### 3.1 ARCHITECTURE

As shown in Fig. 3, CATOK is a diffusion autoencoder with a causal ViT encoder $\mathcal{E}_\delta$ and a MeanFlow DiT decoder $\mathcal{D}_\theta$ parameterized by $\delta$ and $\theta$ respectively. Specifically, given an image $x$, we concatenate it with $K$ registers $R$ and send them into the encoder:

$$[H_e, V_K] = \mathcal{E}_\delta([x, R]),\tag{8}$$

where $H_e$ denotes the image features and $V_K$ represents the compressed 1D tokens. Furthermore, a causal attention mask is applied to enforce the dependency structure among 1D tokens (Cui et al., 2024; Bachmann et al., 2025; Wen et al., 2025). Specifically, image features can attend to each other but not to the 1D tokens; in contrast, 1D tokens are allowed to attend to all image features while being restricted to only their preceding 1D tokens.

In the MeanFlow DiT decoder phase, we first independently sample two timesteps $r$ and $t$, ensuring that $r, t \in [0, 1]$ and $r < t$. Then, the flow path is constructed by linearly interpolating the image $x$ with random noise $\epsilon \sim \mathcal{N}(0, 1)$:

$$z_t = (1 - t)x + t\epsilon.\tag{9}$$

By conditioning the noised image $z_t$ with the 1D tokens from the interval $[r \cdot K, t \cdot K]$, denoted as $V_{r:t}$, and timesteps $r, t$, the DiT decoder predicts the average velocity $u_\theta$ over the time interval:

$$u_\theta = \mathcal{D}_\theta(z_t, r, t, V_{r:t}).\tag{10}$$

Since accurately modeling the instantaneous velocity field improves training stability when learning the average velocity field (Geng et al., 2025; Peng et al., 2025), we follow Eq. (5) and set $r = t$ to model the instantaneous velocity field $v_\theta$:

$$v_\theta = \mathcal{D}_\theta(z_t, t, t, V_K),\tag{11}$$

and all the 1D tokens $V_K$ are conditioned upon.

## 3.2 TRAINING

As illustrated in Fig. 3, CATOK is jointly optimized with two flow objectives—MeanFlow (Geng et al., 2025) and Rectified Flow (Liu et al., 2022; Lipman et al., 2022)—and two representation alignment objectives—REPA (Yu et al., 2024d) and our proposed REPA-A.

**MeanFlow objective.** From Eq. (1), Eq. (7) and Eq. (10), we define our MeanFlow objective as:

$$\mathcal{L}_{MF} := \mathbb{E}||u_\theta - (\epsilon - x) - \text{sg}[(t-r)((\epsilon - x)\partial_z u_\theta + \partial_t u_\theta)]||_2^2, \qquad (12)$$

where $\text{sg}[\cdot]$ denotes the stop-gradient operation, and $(\epsilon - x)\partial_z u_\theta + \partial_t u_\theta$ is computed using the Jacobian-vector product operation.

**Recitified Flow objective.** We also model the instantaneous velocity field to enhance training stability. Based on Eq. (1), we define our Rectified Flow objective as follows:

$$\mathcal{L}_{RF} := \mathbb{E}||v_\theta - (\epsilon - x)||_2^2. \qquad (13)$$

Following Geng et al. (2025), we employ an adaptive $L_2$ loss in place of the standard $L_2$ loss to enhance performance, defined as $\mathcal{L}_{\text{adaptive}} = ||\Delta||_2^2 / \text{sg}[(||\Delta||_2^2 + c)^w]$, where $\Delta$ denotes the regression error, and integrate the two objectives in $\mathcal{L}_F$ by fixing a proportion $q$ of samples with $r = t$. In our implementation, we set $c = 10^{-3}$, $w = 1.0$, and $q = 75\%$.

**REPA objective.** REPA (Yu et al., 2024d) is a regularization technique that leverages Vision Foundation Models (VFMs) to assist DiT training and accelerate convergence. Formally, given the hidden states $H_d$ from a middle layer of the DiT decoder and pretrained representations $H_{vfm}$ from a VFM, our REPA objective can be defined as:

$$\mathcal{L}_{REPA} := -\mathbb{E}[\frac{1}{N}\sum_{n=1}^{N}\text{sim}(H_{vfm}^{[n]}, \text{proj}(H_d^{[n]}))], \qquad (14)$$

where $n$ is a patch index, $\text{sim}(\cdot, \cdot)$ is the cosine similarity function and $\text{proj}(\cdot)$ is the projection layer.

**Our proposed REPA-A objective.** Unlike REPA-E (Leng et al., 2025), which backpropagates gradients to the VAE (Kingma & Welling, 2014), or VA-VAE (Yao et al., 2025), which directly regularizes the compressed features of VAE using VFMs, we propose REPA-A, a representation alignment method specifically tailored for conditional diffusion autoencoders such as our CATOK. Formally, given the image features $H_e$ from the ViT encoder and the same VFM representations $H_{vfm}$, the REPA-A objective can be defined as:

$$\mathcal{L}_{REPA-A} := -\mathbb{E}[\frac{1}{N}\sum_{n=1}^{N}\text{sim}(H_{vfm}^{[n]}, H_e^{[n]})], \qquad (15)$$

where $n$ is a patch index and $\text{sim}(\cdot, \cdot)$ is the cosine similarity function. With REPA-A, the encoder produces higher quality semantic representations, allowing 1D tokens to extract more informative and discriminative visual content, thereby accelerating convergence and enhancing overall performance.

## 3.3 AUTOREGRESSIVE MODELING

Once the causal 1D tokens $V_K$ are obtained from a well-trained CATOK encoder, we train a standard autoregressive model following the "next-token prediction" paradigm to generate images. Formally, the AR model defines the generation process as:

$$p(V_1, V_2, ..., V_k) = \prod_{k=1}^{K} p(V_k|V_1, ..., V_k). \qquad (16)$$

When $V_k$ is represented as discrete indices, this probabilistic model can be optimized via cross-entropy loss. In contrast, when $V_k$ is continuous-valued, as in our setting, optimization is performed using a diffusion loss introduced in Li et al. (2024b). More details are provided in Appendix A.

For image generation, given a prior such as a class token, we first obtain a predicted sequence $\hat{V}_K$ via Eq. (16). By feeding it into MeanFlow decoder, we can directly perform one-step sampling to render an image through $\hat{x} = \epsilon - \mathcal{D}_\theta(\epsilon, 0, 1, \hat{V}_K)$, where $\epsilon$ denotes a random Gaussian noise.

## 4 RELATED WORKS

AR modeling requires compressing raw data into a sequence of tokens, which in turn has spurred a line of research on visual tokenizers. In this section, we categorize them into three types.

**2D visual tokenizers.** VQ-VAE (Van Den Oord et al., 2017; Razavi et al., 2019) is one of the most widely adopted 2D visual tokenizers, integrating Vector Quantization (VQ) into the VAE (Kingma & Welling, 2014) to produce discrete tokens from image patches. Subsequent works improve upon this design: VQGAN (Esser et al., 2021) introduces an adversarial loss to enhance reconstruction quality, while RQ-VAE (Lee et al., 2022) employs multiple quantization stages. MAGVIT-v2 Yu et al. (2024a) further alleviates quantization bottlenecks with Look-up Free Quantization (LFQ), and MaskBit modernizes the VQGAN framework with binary quantized tokens. Most recently, VAR models (Tian et al., 2024; Han et al., 2025) tokenize images into multi-scale 2D tokens and establish a coarse-to-fine ordering via "next-scale prediction". However, these 2D tokenizers either lack causality across tokens or compromise the "next-token prediction" paradigm.

**1D visual tokenizers.** SEED (Cui et al., 2024) employs a causal Q-Former (Li et al., 2023) to extract 1D tokens from a ViT Dosovitskiy et al. (2021) encoder and performs semantic reconstruction with a pre-trained text encoder. TiTok (Yu et al., 2024c) derives 1D tokens using learnable registers and conditions a ViT decoder for mask-to-patch reconstruction. Building on these designs, a line of work explores 1D causal visual tokenizers. TexTok (Zha et al., 2025) and TA-TiTok (Kim et al., 2025) leverage textual conditioning to enhance performance, ALIT (Duggal et al., 2025) introduces adaptive-length tokenization via recurrent encoding, One-D-Piece (Miwa et al., 2025) applies nested dropout (Rippel et al., 2014) on tokens to introduce causality, and SpectralAR (Huang et al., 2025) adopts a similar architecture but imposes explicit spectral interpretations to supervise different tokens. In contrast, CATOK adopts a diffusion-based decoder, which we introduce next.

**Diffusion autoencoders as 1D tokenizers.** Diffusion autoencoders (Preechakul et al., 2022; Yang & Mandt, 2023; Li et al., 2024a; Wang et al., 2025c) compress image features into 1D tokens, which serve as conditioning inputs for diffusion models trained with denoising or rectified flow objectives. However, naïve flow decoders such as FlowMo (Sargent et al., 2025) and DiTo (Chen et al., 2025) condition on all tokens simultaneously, eliminating causal structure and thereby hindering AR learning. To address this, consistency decoders introduce causality through nested dropout, conditioning only on early tokens. The early-token set is determined either stochastically, as in FlexTok (Bachmann et al., 2025) and Semanticist (Wen et al., 2025), or deterministically via timestep binding, as in DDT (Pan et al., 2025) and Selftok (Wang et al., 2025a). However, because earlier tokens are disproportionately favored, these methods induce imbalance, which can degrade AR generation quality. In contrast, our CATOK leverages an additional MeanFlow (Geng et al., 2025) objective to capture causality in a balanced manner while naturally supporting one-step sampling.

## 5 EXPERIMENTS

For fair comparison, we follow common practice (Li et al., 2024b) and conduct experiments on ImageNet-1K (Deng et al., 2009) at $256 \times 256$ resolution.

### 5.1 IMPLEMENTATION DETAILS.

**CATOK.** The CATOK encoder is a ViT-B/8 (Dosovitskiy et al., 2021) with registers (Darcet et al., 2024) and causal attention masks (Cui et al., 2024; Bachmann et al., 2025). For fair comparison, the extracted 1D tokens are 16-dimensional and normalized before being passed to the decoder following (Li et al., 2024a; Wen et al., 2025). The decoder is either a DiT-B/4 or DiT-L/2 (Peebles & Xie, 2023), which are denoted as CATOK-B and CATOK-L respectively, operating on the latent space of a frozen, publicly available KL-16 MAR-VAE (Li et al., 2024b) to reduce computation. Both the encoder and decoder are trained from scratch on ImageNet-1K (Deng et al., 2009) training split. Besides, we utilize DINOv2-B/16 (Oquab et al., 2024) as the VFM of REPA and REPA-A, and the loss weights for $\mathcal{L}_R$, $\mathcal{L}_{REPA}$, and $\mathcal{L}_{REPA-A}$ are set to 1.0, 1.0, and 0.8, respectively.

**Autoregressive modeling.** Following (Wen et al., 2025), we evaluate frozen CATOK by training autoregressive generators LlamaGen (Sun et al., 2024) with a diffusion loss (Li et al., 2024b). The input sequence is conditioned with a learnable class token, which is randomly dropped with

Table 1: **Reconstruction results on ImageNet 256×256 benchmark.** "Token" denotes the number of tokens used for reconstruction, and "Dim." denotes the dimension of these tokens. "Param." indicates the model size, and "VQ" specifies whether the tokens are vector-quantized. "↓" or "↑" denote lower or higher values are better. †: enabling one-step sampling. ∗: without one-step render.

| Method | Token | #Dim. | #Param. | Epochs | VQ | rFID↓ | PSNR↑ | SSIM↑ |
|---|---|---|---|---|---|---|---|---|
| *One-step 2D tokenizers* | | | | | | | | |
| VQGAN | 16x16 | 16 | 307M | - | ✓ | 7.94 | - | - |
| LlamaGen | 16x16 | 8 | 72M | 40 | ✓ | 2.19 | 20.67 | 0.589 |
| MaskBit | 16x16 | 12 | 54M | 270 | ✓ | 1.37 | 21.50 | 0.560 |
| MAR-VAE | 16x16 | 16 | - | - | × | 1.22 | - | - |
| OpenMagViT-V2 | 16x16 | - | 116M | 270 | ✓ | 1.17 | 21.63 | 0.640 |
| *One-step 1D tokenizers* | | | | | | | | |
| SpectralAR-64 | 64 | 16 | 172M | 300 | ✓ | 4.03 | - | - |
| TiTok-S-128 | 128 | 16 | 44M | 300 | ✓ | 1.71 | 17.52 | 0.437 |
| TiTok-L-32 | 32 | 8 | 614M | 300 | ✓ | 2.21 | 15.60 | 0.359 |
| One-D-Piece-B-256 | 256 | 16 | 172M | 300 | ✓ | **1.11** | 18.77 | - |
| CATOK-B-256† | 256 | 16 | 224M | 80 | × | 4.89 | 20.77 | 0.617 |
| CATOK-L-32† | 32 | 16 | 552M | 160 | × | 4.48 | 17.25 | 0.441 |
| CATOK-L-256† | 256 | 16 | 552M | 160 | × | 4.63 | **20.99** | **0.630** |
| *Diffusion tokenizers* | | | | | | | | |
| FlexTok d12-d12 | 256 | 6 | 170M | 640 | ✓ | 4.20 | - | - |
| FlexTok d18-d18 | 256 | 6 | 573M | 640 | ✓ | 1.61 | - | - |
| FlexTok d18-d28 | 256 | 6 | 1.4B | 640 | ✓ | 1.45 | 18.53 | 0.465 |
| Semanticist-L-256 | 256 | 16 | 552M | 400 | × | **0.78** | 21.61 | 0.626 |
| SelfTok-512∗ | 512 | 16 | - | - | ✓ | - | 21.86 | 0.600 |
| FlowMo-Lo-256 | 256 | - | 945M | 130 | ✓ | 0.95 | 22.07 | 0.649 |
| CATOK-B-256 | 256 | 16 | 224M | 80 | × | 1.17 | 22.10 | 0.666 |
| CATOK-L-32 | 32 | 16 | 552M | 160 | × | 2.03 | 17.85 | 0.465 |
| CATOK-L-256 | 256 | 16 | 552M | 160 | × | 0.84 | **22.72** | **0.681** |

probability 0.1 during training to enable Classifier-Free Guidance (CFG). At inference, we adopt a CFG schedule following (Chang et al., 2023; Li et al., 2024b; Wen et al., 2025) without temperature sampling. Additional details are provided in Appendix B.

## 5.2 RECONSTRUCTION

We report reconstruction FID (Heusel et al., 2017) (distributional dissimilarity), PSNR (pixel-wise MSE), and SSIM (Wang et al., 2004) (perceptual similarity) on the ImageNet-1K validation set at $256 \times 256$ resolution. We evaluate three variants: CATOK-B with 256 1D tokens, CATOK-L with 32 tokens and CATOK-L with 256 tokens. The results are compared against state-of-the-art variants with comparable latent spaces and model sizes. As shown in Tab. 1, among diffusion autoencoders, CATOK-L-256 achieves superior PSNR and SSIM, with SSIM significantly outperforming the 945M FlowMo-Lo-256, while also reaching competitive rFID with less than half the training epochs copared with Semanticist-L-256. Remarkably, CATOK-B-256 attains comparable results with only 80 epochs, demonstrating the high training efficiency of CATOK.

Notably, CATOK-L-256† achieves the best PSNR and SSIM among one-step 1D tokenizers, demonstrating its flexibility in sampling: it supports fast one-step sampling while also benefiting from multi-step sampling for improved reconstruction. However, although its rFID surpasses VQGAN (Esser et al., 2021)—a classic 2D tokenizer—by three points, it still lags behind modern tokenizers. This gap arises because those methods rely on more challenging objectives (*e.g.*, GAN (Goodfellow et al., 2014) loss) and complex training recipes (Yu et al., 2024c; Miwa et al., 2025), whereas CATOK is not specifically optimized for one-step sampling but attains comparable results as a byproduct.

Table 2: **Class-conditional generation results on ImageNet-1K 256 × 256 benchmark.** "#Param." denotes the parameters of generator, "Token" and "Step" indicates the number of tokens and steps used for generation, respectively. "↓" or "↑" denote lower or higher values are better.

| Method | Generator | #Param. | Token | Step | gFID↓ | IS↑ |
|---|---|---|---|---|---|---|
| | *2D autoregressive models* | | | | | |
| VQGAN | Tam. Trans. | 1.4B | 256 | 256 | 15.78 | 74.3 |
| RQ-VAE | RQ-Trans. | 3.8B | 256 | 68 | 7.55 | 134.0 |
| Causal MAR | MAR-L | 481M | 256 | 256 | 4.07 | 232.4 |
| LlamaGen | LlamaGen-L | 343M | 256 | 256 | 3.80 | 248.3 |
| VAR | VAR-d16 | 310M | 680 | 10 | 3.30 | 274.4 |
| | *1D masked-prediction models* | | | | | |
| FlowMo-Lo-256 | MaskGiT-L | 227M | 256 | - | 4.30 | 274.0 |
| TiTok-L-32 | MaskGiT-L | 227M | 32 | 8 | 2.77 | 194.0 |
| TiTok-S-128 | MaskGiT-L | 227M | 128 | 64 | 1.97 | 281.8 |
| | *1D autoregressive models* | | | | | |
| FlexTok d12-d12 | AR Trans. | 1.3B | 32 | 32 | 3.83 | - |
| SpectralAR-64 | VAR | 310M | 64 | 64 | 3.02 | 282.2 |
| Semanticist-L-256 | $\epsilon$LlamaGen-L | 343M | 32 | 32 | 2.57 | 260.9 |
| CᴀTᴏᴋ-L-32 | $\epsilon$LlamaGen-L | 343M | 32 | 32 | 3.40 | 288.6 |

Table 3: **Ablation on technique designs.**

(a) **Ablation on training recipe.** FID@$n$: n-step sampling.

| Method | rFID@1 | rFID@25 |
|---|---|---|
| Naïve Decoder ($\mathcal{L}_{RF}$ in Eq. (13)) | 183.69 | 1.81 |
| + $\mathcal{L}_{MF}$ in Eq. (12) | 4.71 | 1.90 |
| + $\mathcal{L}_{REPA}$ in Eq. (14) | 4.31 | 1.71 |
| + $\mathcal{L}_{REPA-A}$ in Eq. (15) | 3.92 | 1.15 |
| + Selecting tokens in $[r, t]$ | 4.89 | 1.17 |

(b) **Ablation on the causality and balance of 1D visual tokens.**

| Select | Token | rFID | gFID |
|---|---|---|---|
| $[r, t]$ | 256 | 1.17 | 4.91 |
| All | 256 | 1.15 | 13.54 |
| First $k$ | 256 | 1.37 | 9.21 |
| First $k$ | 128 | 5.32 | 7.49 |

## 5.3 AUTOREGRESSIVE GENERATION

Following common prective (Li et al., 2024b), we report generation FID and IS (Salimans et al., 2016) (image quality and class diversity) with evaluation suite provided by Dhariwal & Nichol (2021). For fair comparison and efficient training, we train $\epsilon$LlamaGen-L, *i.e.*, standard LlamaGen (Sun et al., 2024) with the diffusion loss (Li et al., 2024b) modified by Wen et al. (2025), for 400 epochs. As shown in Tab. 2, CᴀTᴏᴋ attains comparable gFID and IS scores to the state-of-the-art tokenizers, with far fewer tokenization training epochs (160 *vs.* 300+), demonstrating its capability to learn 1D causal tokens well-suited for standard autoregressive modeling. We present qualitative visualizations in Fig. 4. It is worth noting that training a state-of-the-art visual generative model is computationally expensive and beyond the scope of this work. Instead, we focus on building a 1D tokenizer that captures visual causality and validating its advantages on AR modeling under fair comparison.

## 5.4 ABLATION STUDY

We conduct ablation studies on the smaller CᴀTᴏᴋ-B-256 models, training for 80 epochs on both reconstruction and generation tasks.

**Improved training recipe.** We present a roadmap from the conventional diffusion autoencoder with naïve decoder to our CᴀTᴏᴋ step by step in Tab. 3a. Traditional DiT decoders lack one-step sampling capability, but equipping them with the MeanFlow objective enables reasonable one-step results. Both REPA and REPA-A accelerate convergence and enhance performance. Moreover, optimizing MeanFlow objective on 1D tokens selected from a time interval $[r, t]$ allows the model to learn visual causality, at the cost of a slight performance drop.

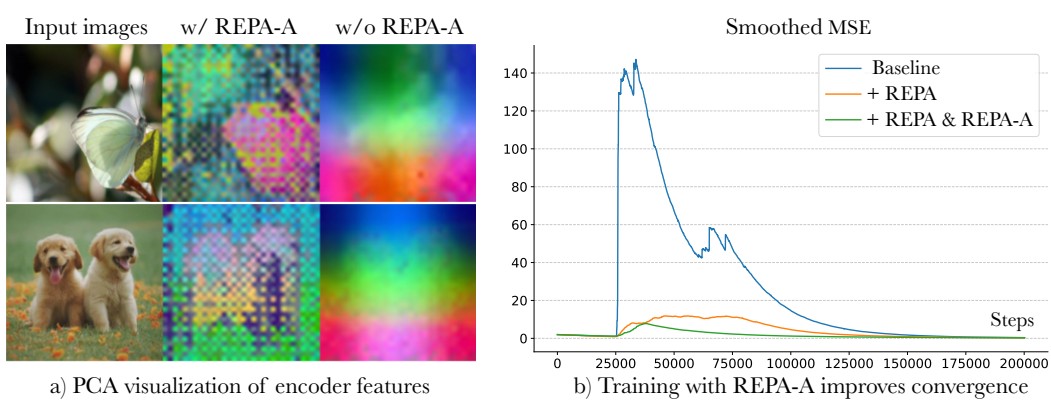

Figure 4: **Qualitative Results.** 256×256 generated images on ImageNet-1K with CATOK-L-32.

Input images   w/ REPA-A   w/o REPA-A                    Smoothed MSE

a) PCA visualization of encoder features          b) Training with REPA-A improves convergence

Figure 5: **Effectiveness of our REPA-A.** a) We apply principal component analysis (PCA) to visualize image features from the CATOK encoder. b) Training curves of the smoothed MSE between prediction and target, with the MeanFlow loss ($\mathcal{L}_{MF}$) added at 25K steps.

**Causality and balance matter in AR modeling.** We evaluate three variants of 1D token selection: (1) selecting tokens within an interval $[r, t]$ (our default setting); (2) selecting all tokens; and (3) selecting the first $k$ tokens. For the third variant, we train two AR models using either all 256 tokens or only the first 128 tokens. As shown in Tab. 3b, CATOK achieves the best gFID. Non-causal tokens hinder AR modeling, and, consistent with Bachmann et al. (2025); Wen et al. (2025), imbalance reduces the contribution of later tokens—an issue that CATOK fundamentally addresses without requiring additional re-weighting mechanism (Wang et al., 2025a).

**REPA-A stabilizes training and improves performance.** As shown in Fig. 5 a), REPA-A makes encoder features more informative and discriminative, helping the registers capture richer content. In Fig. 5 b), REPA-A mitigates the loss spike at 25K steps when the MeanFlow loss is introduced, stabilizing decoder training and improving overall performance.

## 6 CONCLUSION

We presented CATOK, a novel 1D causal image tokenizer to bridge the gap between autoregressive language models and vision models. By binding the average velocity field in the MeanFlow objective to the corresponding 1D token segments, we enabled the diffusion autoencoder to learn visual causality along the flow path while supporting one-step sampling. Furthermore, we proposed an advanced regularization method REPA-A, which effectively stabilized and accelerated the training of the autoencoder. Experiments demonstrated that we achieved state-of-the-art PSNR and SSIM on ImageNet reconstruction, and obtaining comparable results on the class-conditional generation.

## REPRODUCIBILITY STATEMENT

We provide hyperparameter details in Sec. 5 and Appendix B. We will also release the codebase and model checkpoints to reproduce the results in the paper.

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

## A ARCHITECTURE OF AR MODELING

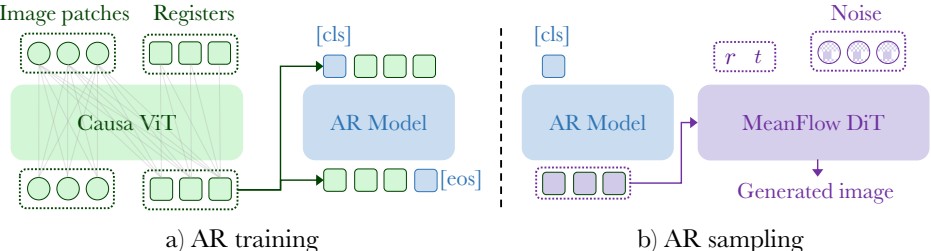

a) AR training                    b) AR sampling

Figure 6: **Architecture of AR modeling.**

We illustrate AR training and sampling in Fig. 6, and visualize the causal mask mechanism in ViT in Fig. 6 a). After training CATOK, we freeze the encoder to extract 1D tokens. During AR training stage, these tokens are optimized with a class token prefix using teacher forcing under a diffusion loss (Li et al., 2024b). At sampling time, we input a learned class token, the AR model predicts the corresponding visual 1D tokens, and these tokens are then conditioned to the decoder for generation.

## B MORE IMPLEMENTATION DETAILS

Table 4: Detailed configuration of CATOK-B and CATOK-L for tokenization and AR modeling.

| Training Config | CATOK-B | CATOK-L | AR modeling |
|---|---|---|---|
| Optimizer | | AdamW | |
| Peak learning rate | | $1 \times 10^{-4}$ | $5 \times 10^{-5}$ |
| Minimum learning rate | | 0 | |
| Learning rate schedule | | cosine decay | constant |
| Batch size | | 1024 | 2048 |
| Weight decay | | 0.05 | |
| Epochs | 80 | 160 | 400 |
| Warmup epochs | | 0 | 96 |
| Gradient clipping | | 3.0 | |
| EMA | | 0.999 | |

Training setup follows Wen et al. (2025), with detailed hyperparameters in Tab. 4. For reconstruction, we disable CFG in one-step sampling, and apply CFG with a scale of 2.0 in 25-step sampling. For 80-epoch training, we introduced the MeanFlow objective at epoch 10 and the selecting mechanism at epoch 40; for 160-epoch training, these corresponded to epochs 20 and 80, respectively. For generation, we do not use CFG with CATOK, and the CFG of AR model is the same as MUSE (Chang et al., 2023), MAR (Li et al., 2024b) and Semanticist (Wen et al., 2025), whici tunes down the guidance scale of small-indexed tokens to improve the diversity of generated sample.

## C THE USE OF LARGE LANGUAGE MODELS (LLMS)

LLMs are utilized for language refinement and are not involved in any other component of this work.

