# OpenReview forum: "CaTok: Taming Mean Flows for One-Dimensional Causal Image Tokenization"
_ICLR.cc/2026/Conference — ICLR 2026 Conference Withdrawn Submission_

### Official Review · Reviewer_bEVS · 2025-10-22

**Soundness:** 2
**Presentation:** 2
**Contribution:** 2
**Rating:** 4
**Confidence:** 4

**Summary:**

This paper presents CaTok, a 1D causal image tokenizer that utilizes a MeanFlow-based diffusion autoencoder for autoregressive image modeling. By introducing a token selection strategy over time intervals and aligning encoder features with representations from Vision Foundation Models (VFMs) using the proposed REPA-A regularization, CaTok captures visual causality, supports both fast one-step and high-fidelity multi-step sampling, and attains competitive results for image reconstruction and autoregressive generation on ImageNet-1k.

**Strengths:**

1. CaTok introduces a principled way to obtain causal 1D tokens from images by coupling a causal Vision Transformer encoder with a MeanFlow diffusion-based decoder, offering a practical approach to image tokenization that aligns well with next-token prediction patterns of language models.

2. The introduction of the REPA-A regularization, which aligns encoder representations with Vision Foundation Models, demonstrably accelerates and stabilizes training dynamics (Fig. 5b). Empirically, this leads to significantly reduced training epochs while maintaining competitive results (Table 1).

3. The ablation studies (Table 3) are thorough, dissecting every major proposed component’s effect on performance, including losses, regularizations, and token selection.

**Weaknesses:**

1. As shown in Table 1, the rFID of CaTok with one-step sampling is worse than the VQ baselines (e.g., One-D-Piece-B-256) with larger parameter size, which questions the effectiveness of this approach.

2. The experiment of image generation in Table 2 is insufficient. It would be better also to provide the performance of CaTok-B-256 and CaTok-L-256, as well as other baselines, to demonstrate the scaling potential of CaTok.

**Questions:**

1. As mentioned in L316-319,  what is the exact role of KL-16 MAR VAE here? Does the latent output of the CaTok decoder need to be further converted into a pixel image through this VAE? If so, what is the actual wall time and FLOPs during training and inference of CaTok for generation and reconstruction compared to baselines?

---

### Official Review · Reviewer_xuWy · 2025-10-31

**Soundness:** 3
**Presentation:** 3
**Contribution:** 3
**Rating:** 4
**Confidence:** 4

**Summary:**

This paper proposing CATOK, a 1D causal image tokenizer paired with a MeanFlow decoder. The key idea is to condition the decoder only on tokens within a sampled time interval
[r,t] and tie that conditioning to the MeanFlow objective so the model explicitly learns the average velocity along the subpath. This is positioned as a remedy for two issues in prior diffusion autoencoders: (i) “naïve” decoders that condition on all tokens (no causality), and (ii) “consistency/nested-dropout” decoders that bias toward early tokens (imbalance). The system adds REPA-A, a representation alignment loss that matches encoder features to a vision foundation model to stabilize and speed training. Empirically, the authors claim strong ImageNet reconstructions (22.72 PSNR / 0.681 SSIM) and show that the AR head trained on CATOK’s tokens reaches competitive gFID/IS, with the added perk of one-step as well as multi-step sampling. The architecture and training objectives are clearly laid out and ablations indicate that interval-based conditioning improves balance/causality versus “first-k” or “all-tokens” baselines.

The novelty feels incremental: the approach leans heavily on a recently introduced MeanFlow objective and largely reframes causality as interval selection plus conditioning, with REPA-A supplying extra supervision via external VFMs. This raises questions about where the wins truly come from—MeanFlow vs. token interval selection vs. VFM alignment—and whether REPA-A is simply injecting semantic priors that confound comparisons to tokenizers trained without such help. The evidence for “causality” is mostly qualitative (fine-to-coarse trends and segment reconstructions), while the AR evaluation is modest (e.g., 400-epoch training and “comparable” results rather than clear SOTA), and ablations admit a slight performance drop when enforcing interval conditioning. Important practical aspects remain underexplored: scalability to higher resolutions, text-conditional generation, robustness beyond ImageNet, and apples-to-apples cost/quality trade-offs versus strong 2D/VAR tokenizers. Overall, it’s a neat, well-engineered recipe with promising sampling behavior, but the causal claim and comparative gains need sharper, more controlled evidence to justify publication at a top venue.

**Strengths:**

The step-by-step roadmap (MeanFlow → REPA → REPA-A → interval token selection) shows measurable gains and isolates each component’s effect on rFID.

**Weaknesses:**

1. All experiments are on ImageNet-1K at 256×256; no evidence for higher resolutions, other datasets, or broader generalization.

2. Support rests mainly on ablations and qualitative trends; even the authors note interval conditioning brings a performance drop, which undercuts the claim that causality is unequivocally beneficial.

3. The method combines MeanFlow, Rectified Flow (with adaptive L2 and r=t mixing), plus two representation-alignment losses—nontrivial to reproduce and tune.

4. The AR head is trained 400 epochs and framed as “comparable,” which is a modest bar and leaves headroom versus current best systems.

**Questions:**

How do you operationally verify “causality”? Beyond qualitative trends, is there a quantitative test showing information flows strictly from earlier to later tokens?

Can you disentangle gains from MeanFlow vs. interval conditioning vs. REPA-A? (e.g., factorial ablation with matched training budgets and early-stopping rules.)

How robust are results to the upstream VAE choice (MAR-VAE vs. other latents)? Any performance collapse or re-tuning required when swapping latents?

---

### Official Review · Reviewer_3aMS · 2025-11-01

**Soundness:** 3
**Presentation:** 3
**Contribution:** 3
**Rating:** 4
**Confidence:** 4

**Summary:**

This paper presents CaTok, a one-dimensional causal image tokenizer based on diffusion autoencoders equipped with a MeanFlow decoder. The key insight is to bring causality into visual tokenization, aligning image token learning with the autoregressive “next-token prediction” paradigm of language models. CaTok conditions the decoder on 1D token intervals [r,t], modeling the average velocity field along subpaths in the diffusion process to preserve temporal causality and token balance. In addition, the authors propose REPA-A, a representation alignment regularization that leverages Vision Foundation Models to stabilize and accelerate training. Experiments on ImageNet-1K (256×256) show that CaTok achieves high PSNR and SSIM in reconstruction, supports both one-step and multi-step sampling, and produces competitive generative performance compared to prior 1D and 2D tokenizers.

**Strengths:**

1. Using the MeanFlow objective to model average velocity over intervals [r,t] is a clever way to enforce causal consistency while allowing one-step sampling and balanced token usage.

2. The addition of REPA-A provides both empirical and conceptual contributions, leading to faster convergence and better feature quality.

3. CaTok achieves competitive or better performance with significantly fewer training epochs than comparable models.

**Weaknesses:**

1. The motivation is not well-described. The paper does not clearly justify why enforcing stronger causality in visual tokenization is inherently beneficial. While the motivation draws inspiration from autoregressive language models, the authors do not provide theoretical or empirical evidence showing that causal dependencies are necessary—or even advantageous—for visual representations, which may naturally rely more on spatial coherence than temporal order.

2. The model is only evaluated on ImageNet-256. Can it generalize to higher resolutions such as 512*512?

3. The proposed model obtain good PSNR and SSIM results, yet falls short in FID scores. What leads to this inconsistency?

**Questions:**

See Weaknesses.

---

### Official Review · Reviewer_SQ59 · 2025-11-01

**Soundness:** 3
**Presentation:** 3
**Contribution:** 2
**Rating:** 4
**Confidence:** 3

**Summary:**

This paper presents CaTok, a 1D causal image tokenizer that addresses the causality vs balance trade-off in diffusion autoencoders through a MeanFlow decoder design. Previous approaches either lack causality or introduce imbalance between tokens, and CaTok's design allows interval-based token conditioning and also enables one-step generation. Empirical results support the validness of this design.

**Strengths:**

- This paper identifies that previous approaches either lack causality or introduce imbalance between tokens, which is interesting and inspiring. The proposed method CaTok allows interval-based token conditioning and enables one-step generation, which are meaningful properties for research in this line.
- The paper clearly articulates the problem, solution, and experimental validation. The comparison framework (Figure 2) effectively illustrates the key differences between approaches.

**Weaknesses:**

- Some technical contributions, for instance employing mean flow for diffusion autoencoders and the use of REPA-A, might be incremental. It is plausible to use them but hardly claimed as a major contribution.
- The performance on generation benchmarks is modest compared to the closest counterpart Semanticist. Was that because the tokenizer was not fully trained, and how would it perform if trained for 400 epochs? Also, there is a large gap between the reconstruction performance of CaTok-L-32 and CaTok-L-256, but generation performance of CaTok-L-256 was not reported.
- Can the authors elaborate or justify more on the benefits of interval-based token conditioning?

**Questions:**

see weaknesses

---

### Note · Authors · 2025-11-14

I have read and agree with the venue's withdrawal policy on behalf of myself and my co-authors.